# Perceived stress, stigma, and social support among Nepali health care workers during COVID-19 pandemic: A cross-sectional web-based survey

Lok Mani Giri[1][☯], Kiran Paudel[2,3][☯]*, Sandesh Bhusal[2,3], Tara Ballav Adhikari[3,4,5], Gabriel Gulis[1]

1 Unit for Health Promotion Research, University of Southern Denmark, Esbjerg, Denmark, 2 Institute of Medicine, Tribhuvan University, Maharajgunj, Nepal, 3 Nepal Health Frontiers, Tokha, Kathmandu, Nepal, 4 COBIN Project, Nepal Development Society, Bharatpur, Chitwan, Nepal, 5 Department of Public Health, Section for Global Health, Aarhus University, Aarhus, Denmark

☯ These authors contributed equally to this work.
* kiranpaudel59@gmail.com

**Data Availability Statement:** The raw data in the form of tables has been uploaded as supporting information.

## Abstract

The COVID-19 pandemic has caused immense psychological distress among Health Care Workers (HCWs). HCWs have been stigmatized by people at their workplace and community, fearing that health care workers are the sources of spreading the virus. This study aimed to assess the perceived stress, stigma, and social support of Nepali health care workers during the COVID-19 pandemic. A web-based cross-sectional survey was conducted among 380 Nepali HCWs. Perceived stress was measured using Perceived Stress Scale (PSS-10). The questionnaire related to stigma was derived from a study conducted to measure stigma among HCWs during the Middle East respiratory syndrome coronavirus (MERS-CoV) outbreak. Furthermore, perceived social support was measured by Oslo Social Support Scale (OSSS-3). Associated factors were examined using Chi-square tests followed by multivariate logistic regression analyses at the significance level of 0.05. This study illustrated that nearly half (44.2%) of the respondents perceived poor social support. Similarly, almost 3% of the HCWs experienced high perceived stress, whereas 87.6% perceived moderate stress, and 35% of the HCWs felt stigmatized. Nepali healthcare workers experienced perceived stress, social stigma, and social support in various severity levels during the COVID-19 pandemic. COVID-19 emergency is emotionally difficult and where psychological distress can jeopardize outcomes and affect work performance. Appropriate psychological interventions are needed to promote the mental well-being of the healthcare workers.

**Funding:** The author(s) received no specific funding for this work.

**Competing interests:** The authors have declared that no competing interests exist.

## Introduction

The unprecedented spread of Corona Virus Disease 2019 (COVID-19) has caused a global public health crisis. Nepal is no exception to this pandemic with overwhelming effects on its economy and healthcare system [1]. As of 7th March 2022, 977567 cases and 11948 deaths had been reported in the country despite adopting operative measures like nationwide lockdown, social distancing, and travel restrictions [2].

The weaker health system capacity in terms of finance, medical products, technology, and human resources for health created extreme pressure for providing appropriate diagnostic and treatment services; due to which the working life of health service providers has become more stressful than normal [3]. Evidence around the world has shown that health care workers (HCWs) directly involved in the diagnosis, treatment, and care of patients with COVID-19 were at the risk of developing mental health problems [4–6].

HCWs in the pandemic are more vulnerable to emotional distress as they have a greater risk of exposure to the virus, increased workload, fear of infecting their family and friends, lack of experience in managing the disease, and lack of personal protective equipment (PPE) [7–9]. Apart from this, perceived stigma and social discrimination contributed to significant mental problems among HCWs [10, 11]. The mental health impact of the outbreaks was usually overshadowed during emergencies due to a highly stressful environment and hard to manage in an ordinary way, although the consequences of this negligence are costly [12].

The role of social support and its protective relationship to mental health difficulties has widely been focused on the current pandemic. A strong and statistically validated relationship between poorer mental health and a low level of perceived social support was reported by various studies conducted among HCWs [13, 14].

A comprehensive understanding of the psychological burden among different groups of health service providers is essential to design an appropriate psychological support system that could promote the mental well-being of HCWs and align them to the patient's needs [15, 16]. During the COVID-19 pandemic, sleep problems, anxiety, and depression appeared to be common and were linked to higher levels of psychological distress [17, 18]. Anxiety symptoms and fear of COVID-19 were the common predictive factors of distress and sleep problems [19]. Due to the lack of medicine and vaccination for COVID-19, in the beginning, people stigmatized front-line health care workers, which may have resulted in self-stigmatization and mental health problems. During the COVID-19 pandemic, mental health services like early identification and treatment of problems among health care personnel would motivate them to serve efficiently and safely. Aside from psychological distress, our research will aid in identifying possible indicators of social support and stigma that may impact the physical and mental health of Nepali health care workers. Previous studies conducted in Nepal did not look into stigma, social support, and stress despite the many studies in Nepal included mental health outcomes and fears [3, 12, 20]. There is scant information about stigma and social support relating to stress, which could be an interesting outcome to look at in low and middle-income countries. A timely assessment of the stress, stigma, and social support during emergencies will help the management to respond and reduce psychological distress and help make the public aware of the harmful impact of stigmatization and the importance of social support during such humanitarian situations in the future and in such pandemic. Therefore, this research aims to assess the perceived stress, stigma, and social support among Nepali HCWs during the COVID-19 pandemic.

## Methods

### Study design and study population

A web-based cross-sectional study was conducted among health care workers in Nepal. The study population was the Nepali healthcare workers working during the COVID-19 pandemic in different healthcare facilities across all seven provinces. The healthcare facilities included the government hospitals, private hospitals, academic institutes/medical colleges, and fever clinics set up during the pandemic. Similarly, healthcare workers included doctors, nurses, pharmacists, diagnostic personnel, paramedics, and public health practitioners. The health workers aged 18 years and above and currently living in Nepal and working in the COVID-19 management were eligible for the study.

### Sampling

The expected proportion of perceived stress of COVID-19 among healthcare providers was taken as 56% from a similar study conducted in Oman [21] and the sample size was calculated using the formula; $n = z^2 pq/d^2$, where p = 0.56, q = 0.44, z = 1.96 at 95% confidence interval, and d = 0.05 is 379. After assuming 10% as the non-response rate, the final calculated sample size was 422.

### Data collection

Online Google forms questionnaire was administered to the participants through social media platforms to collect the data following the non-probability sampling method. To limit non-health worker's responses to the online survey, forms were only posted in the relevant official social media groups of health workers such as the doctor society of Nepal, registered nurses Nepal, Government doctors association of Nepal, Nepali doctors lounge, Nepali health professional, etc.

### Measures

Perceived stress was assessed using the Perceived Stress Scale (PSS-10). The PSS-10 consists of six positively (Items 1, 2, 3, 6, 9, and 10: Positive factor) and four negatively (Items 4, 5, 7, and 8: Negative factor) worded items. Individual scores on the PSS range from 0 to 40, with higher scores indicating higher perceived stress. For the bivariate analysis, we used the mean score of the sample as the cut-off value to distinguish between low stress and high stress (>20).

The questionnaire related to stigma among healthcare workers was based on a study used to measure stigma among nursing staff during the Middle East respiratory syndrome coronavirus (MERS-CoV) outbreak [22]. The stigma scale comprises 13 items; each scored on a 5-point Likert scale. The total score ranges between 0 and 52, with a higher score indicating that the HCWs perceived greater stigma. Further analysis, a mean cut-off score of 26 was used to categorize stigma [23].

Perceived Social Support was measured using the Oslo Social Support Scale (OSSS-3). The sum scores of this scale range from 3 to 14, with a higher value representing a strong level and lower values representing a poor level of social support. Social support levels are broadly categorized as "poor support" 3–8, "moderate support" 9–11, and "strong support" 12–14 [24]. For the bivariate analysis, moderate and strong social support groups were combined to form a single group.

Socio-demographic characteristics included age, gender, marital status, family type, monthly income, etc., and work-related variables included; work experience, profession, type

of health facility, COVID-19 status, vaccination status, etc. Variables and their definition used in the study are tabulated. (Table 1)

## Data analysis

After excluding the 24 samples as they were not currently residing in Nepal, the final 380 responses were eligible for the analysis. Statistical analysis was performed using IBM SPSS Statistics for Windows, Version 26 (IBM SPSS Statistics for Windows, IBM Corporation, Armonk, NY).

Descriptive analysis of the variables was done in terms of frequency and percentage. The chi-Square test was used to determine the association between the dependent variable (perceived stress, stigma, and social support) and independent variables (socio-demographic factors, work-related variables) which is shown in S1–S3 Tables. Variables with a p-value less than 0.1 during bivariate analysis were entered into the regression model [25]. The adjusted odds ratio was calculated at a 95% confidence interval (CI), and a p-value less than 0.05 was considered statistically significant.

## Ethical considerations

Ethical approval was obtained from the Ethical Review Board of Nepal Health Research Council, (Reference number: 1900). All the respondents were informed about the aims and objectives of the study by including the written consent form in the questionnaire itself. Written consent was taken from study participants prior to completing the survey form. Participants

**Table 1. List of study variables.**

| S. N. | Variables | Categories of variables |
|---|---|---|
| | **Socio-demographic variables** | |
| 1 | Age | Less than 30 years, 30–45 years, 45 and above |
| 2 | Gender | Male, Female |
| 3 | Marital status | Currently Unmarried, Currently Married |
| 4 | Type of family | Joint/Extended(Family consisting of others like grandparents, siblings, etc. including parents and their children) |
| | | Nuclear (Family consisting of parents and their children only) |
| 5 | Currently working provinces | Province 1, Province 2, Bagmati Province, Gandaki Province, Lumbini Province, Karnali Province, Sudurpaschim Province |
| 6 | Currently staying | In hostel/quarter, In own home/rented house |
| 7 | Average monthly income | Below Nepali Rupee (Rs). 40,000 (USD 326), Above Rs. 40,000 (USD 326) |
| | **Work-related variables** | |
| 1 | Job experience | Less than 5 years, 5 to 10 years, 10 years and above |
| 2 | Stayed in isolation | Yes, No |
| 3 | Health Professional | Doctor, Lab personnel, Nurse, Paramedics, Radiological Professional |
| 4 | Type of hospital currently working | Medical college/academy, Private Hospital, Public Hospital, Other facilities with COVID19 clinic |
| 5 | Staying away from your family | Yes, No |
| 6 | Got training/orientation regarding COVID19 | Yes, No |
| 7 | Receive the vaccine for COVID19 | Yes, No |
| 8 | Get infected by the COVID19 | Yes, No |

gave their consent by ticking the designated box. No personal identities were collected during the study to ensure their confidentiality.

# Results

## Characteristics of the study participants

A total 380 participants were eligible for the analysis. The median age (in years) of the study participants was 35 years. Of the study participants, 58.2% were male. Most of the healthcare workers were paramedics (26.8%), followed by nurses (24.7%), and medical doctors (22.6%).

More than half of the participants had a family income below Rs. 40,000. 54.5% of the participants were staying away from their homes. Similarly, the majority of the participants (48.2%) were currently working at the public hospital and 17.4% were infected with COVID-19. The orientation/training of the COVID-19 pandemic was received by 58.7% and 69.7% of the participants had received the first dose of vaccine against COVID-19 (Table 2).

## Level of perceived stress, stigma, and social support

Table 3 illustrates the level of perceived stress, stigma, and social support among Nepali HCWs during the COVID-19 pandemic. Nearly 3% of the HCWs experienced high perceived stress but most of the HCWs (87.6%) had moderate perceived stress and 35% of the HCWs felt stigmatized. Similarly, 44.2% of the respondents perceived poor social support and only 5.5% of them felt to have strong social support.

## Associated factors of perceived stress, stigma, and social support

The logistic regression analysis presented in Table 4 describes that female HCWs had higher odds of perceived stress than male HCWs (AOR = 2.4, 95% CI: 1.3–4.3). Similarly, the strongest effect size was observed among professions, lower odds of perceived stress was found among medical doctors (AOR = 0.2, 95% CI: 0.1–0.5) compared to radiological personnel. Age, marital status, average monthly income, and working experiences of the respondents were also significantly associated with perceived stress but after adjustment, the relationship was attenuated and became non-significant.

HCWs from nuclear families were 1.7 times more likely (AOR = 1.7, 95% CI: 1.0–2.9) to be stigmatized in comparison to those who were from joint/extended families. Similarly, the HCWs currently working at Bagmati and Karnali Province were 3.6 and 2.9 times more likely to be stigmatized than those working in Sudhurpaschim Province. The respondents from the

**Table 2. Characteristics of the study participants.**

| Characteristics | Number | Percentage |
|---|---|---|
| **Age (in years)** | | |
| Less than 30 years | 96 | 25.3 |
| 30–45 years | 257 | 67.7 |
| 45 years and above | 27 | 7.1 |
| **Gender** | | |
| Male | 221 | 58.2 |
| Female | 159 | 41.8 |
| **Marital Status** | | |
| Currently Unmarried | 133 | 35 |
| Currently Married | 247 | 65 |

*(Continued)*

**Table 2.** (Continued)

| Characteristics | Number | Percentage |
|---|---:|---:|
| **Type of Family** | | |
| Joint/Extended | 223 | 58.7 |
| Nuclear | 157 | 41.3 |
| **Currently working provinces** | | |
| Province 1 | 28 | 7.4 |
| Province 2 | 54 | 14.2 |
| Bagmati | 98 | 25.8 |
| Gandaki | 46 | 12.1 |
| Lumbini | 71 | 18.7 |
| Karnali | 40 | 10.5 |
| Sudhurpaschim | 43 | 11.3 |
| **Currently staying** | | |
| In hostel/quarter | 109 | 28.7 |
| In own home/rented house | 271 | 71.3 |
| **Average monthly income** | | |
| Below Rs.40,000 (USD 326) | 229 | 60.3 |
| $\geq$ Rs. 40,000 (USD 326) | 151 | 39.7 |
| **Job experience** | | |
| Less than 5 years | 147 | 38.7 |
| 5 to 10 years | 133 | 35 |
| 10 years and above | 100 | 26.3 |
| **Stayed in isolation** | | |
| Yes | 116 | 30.5 |
| No | 264 | 69.5 |
| **Health Professional** | | |
| Doctor | 86 | 22.6 |
| Lab. personnel | 65 | 17.1 |
| Nurse | 94 | 24.7 |
| Paramedics | 102 | 26.8 |
| Radiological personnel | 33 | 8.7 |
| **Type of health facility currently working** | | |
| Medical college/academy | 66 | 17.4 |
| Private hospital | 90 | 23.7 |
| Public hospital | 183 | 48.2 |
| Other facilities with COVID-19 clinic | 41 | 10.8 |
| **Staying away from your family** | | |
| Yes | 207 | 54.5 |
| No | 173 | 45.5 |
| **Got training/orientation on COVID-19** | | |
| Yes | 157 | 41.3 |
| No | 223 | 58.7 |
| **Received the first dose of COVID-19 vaccine** | | |
| Yes | 265 | 69.7 |
| No | 115 | 30.3 |
| **Infected by the COVID-19** | | |
| Yes | 66 | 17.4 |
| No | 314 | 82.6 |

**Table 3. Level of perceived stress, stigma, and social support.**

| Variables | Categories | Number | Percentage |
|---|---|---|---|
| **Perceived Stress** | Low | 36 | 9.5 |
| | Moderate | 333 | 87.6 |
| | High | 11 | 2.9 |
| **Stigma** | Stigmatized | 133 | 35.0 |
| | Non stigmatized | 247 | 65.0 |
| **Social Support** | Poor | 168 | 44.2 |
| | Moderate | 191 | 50.3 |
| | Strong | 21 | 5.5 |

public hospital had (AOR: 0.4, 95% CI: 0.2–0.8) a lower chance of being stigmatized than those working at fever clinics (Table 5).

HCWs living in their own house had two folds (AOR = 2.1, 95% CI: 1.2–3.6) a higher likelihood of perceived social support than those who lived in a hostel/rented house/quarter. Lower odds of social support was found among those who had received the training or orientation of

**Table 4. Association of perceived stress with independents variables.**

| Variables | Perceived stress | | Unadjusted OR (95% CI) | Adjusted OR (95% CI) |
|---|---|---|---|---|
| | Low (%) | High (%) | | |
| **Gender** | | | | |
| Male | 135(61.1) | 86(38.9) | Ref. | Ref. |
| Female | 54(34.0) | 105(66.0) | 3.1 (1.9–4.7) *** | 2.4 (1.3–4.3) * |
| **Age** | | | | |
| 45 years and above | 18(66.7) | 9(33.3) | Ref. | Ref. |
| 30–45 years | 135(52.5) | 122(47.5) | 1.8(0.8–4.2) | 1.6 (0.6–4.2) |
| Less than 30 years | 36(37.5) | 60(62.5) | 3.3(1.4–8.2) * | 1.3 (0.4–4.3) |
| **Marital status** | | | | |
| Currently unmarried | 50(37.6) | 83(62.4) | Ref. | Ref. |
| Currently married | 139(56.3) | 108(43.7) | 0.5 (0.3–0.7) ** | 0.6 (0.3–1.1) |
| **Average monthly income** | | | | |
| Above NRs. 40,000 (USD 326) | 85(56.3) | 66(43.7) | Ref. | Ref. |
| Below NRs. 40,000 (USD 326) | 104(45.4) | 125(54.6) | 1.6(1.0–2.3) * | 0.7(0.4–1.3) |
| **Working Experience** | | | | |
| 10 years and above | 60 (60) | 40 (40) | Ref. | Ref. |
| 5 to 10 years | 74 (55.6) | 59 (44.4) | 1.2(0.7–2.0) | 0.9 (0.5–1.8) |
| Less than 5 years | 55 (37.4) | 92 (62.6) | 2.5(1.5–4.2) * | 1.9 (0.9–4.0) |
| **Health Professional** | | | | |
| Radiological Professional | 12(36.4) | 21(63.6) | Ref. | Ref. |
| Paramedics | 60(58.8) | 42(41.2) | 0.4(0.2–0.9) * | 0.5(0.2–1.0) |
| Nurse | 30(31.9) | 64(68.1) | 1.2(0.5–2.8) | 0.6(0.2–1.6) |
| Medical Laboratory | 28 (43.1) | 37 (56.9) | 0.8 (0.3–1.8) | 0.7 (0.3–1.7) |
| Doctor | 59(68.6) | 27(31.4) | 0.3(0.1–0.6) * | 0.2(0.1–0.5) ** |

*Significant at p <0.05

** Significant at p <0.01

*** Significant at p <0.001

**Table 5. Association of stigma with independents variables.**

| Variables | Stigmatized (%) | Unadjusted OR (95% CI) | Adjusted OR (95% CI) |
|---|---|---|---|
| **Gender** | | | |
| Male | 91(68.4) | Ref. | Ref. |
| Female | 42(31.6) | 0.5 (0.3–0.8) ** | 0.8(0.5–1.4) |
| **Type of family** | | | |
| Joint/Extended | 60(45.1) | Ref. | Ref. |
| Nuclear | 73(54.9) | 0.4 (0.3–0.7) *** | 1.7(1.0–2.9) * |
| **Currently working provinces** | | | |
| Province 1 | 5 (3.8) | 0.8 (0.2–2.8) | 0.7(0.3–2.5) |
| Province 2 | 8 (6.0) | 0.7(0.2–1.9) | 0.7(0.2–2.1) |
| Bagmati | 56 (42.1) | 5.0(2.2–11.6) *** | 3.6(1.4–9.2) * |
| Gandaki | 5 (3.8) | 0.5(0.1–1.5) | 0.4(0.1–1.5) |
| Lumbini | 39 (29.3) | 1.4(0.5–3.9) | 1.1(0.4–3.4) |
| Karnali | 11 (8.3) | 4.6(1.9–10.9) * | 2.9(1.1–7.4) * |
| Sudhurpaschim | 9 (6.8) | Ref. | Ref. |
| **Currently staying** | | | |
| In own home | 64 (48.1) | 0.6(0.4–0.9) * | 1.1(0.6–1.9) |
| In hostel/quarter/rented home | 69 (51.9) | Ref | Ref |
| **Working experience** | | | |
| 10 years and above | 47 (35.3) | Ref | Ref |
| 5 to 10 years | 37 (27.8) | 0.4(0.3–0.8) * | 0.8(0.4–1.5) |
| Less than 5 years | 49 (36.8) | 0.6(0.3–0.9) * | 1.1(0.5–2.0) |
| **Type of hospital currently working** | | | |
| Other facility with covid19 clinic | 18 (13.5) | Ref | Ref |
| Public Hospital | 77 (57.9) | 0.4(0.2–0.7) * | 0.4(0.2–0.8) * |
| Private Hospital | 19 (14.3) | 1.1(0.5–2.1) | 1.2(0.5–2.8) |
| Medical college/academy | 19 (14.3) | 0.6(0.3–1.0) | 0.8(0.4–1.8) |
| **Staying away from your family** | | | |
| Yes | 63 (30.4) | 1.6(1.0–2.4) * | 1.2(0.7–2.2) |
| No | 70 (40.5) | Ref | Ref |
| **Got training/orientation regarding covid-19** | | | |
| Yes | 67 (50.4) | 0.6(0.3–0.9) * | 0.8(0.5–1.4) |
| No | 66 (49.6) | Ref | Ref |

*Significant at p <0.05

** Significant at p <0.01

*** Significant at p <0.001

the COVID-19 pandemic (AOR = 0.5. 95% CI: 0.3–0.8) than those who were not trained or oriented (Table 6).

## Discussion

Our study found that almost 3% of the HCWs experienced high perceived stress, but 87.6% of the HCWs experienced moderate perceived stress. Similarly, 44.2% of them perceived lower social support, and 35.0% of the HCWs responded as being stigmatized. These findings are less than the result shown by the study conducted in the early phase of the pandemic among the HCWs in Nepal [12], Italy [26], and Singapore [27].

**Table 6. Association of social support with independent variables.**

| Variables | Social support | | Unadjusted OR (95% CI) | Adjusted OR (95% CI) |
|---|---|---|---|---|
| | Poor (%) | Moderate/strong (%) | | |
| **Gender** | | | | |
| Male | 89(40.3) | 132 (59.7) | Ref | Ref |
| Female | 79(49.7) | 80 (50.3) | 0.7 (0.5–1.0) | 0.8 (0.4–1.4) |
| **Type of family** | | | | |
| Joint/Extended | 114(51.1) | 109(48.9) | Ref | Ref |
| Nuclear | 54(34.4) | 103(65.6) | 1.9(1.3–3.0) ** | 1.5 (0.9–2.5) |
| **Currently staying** | | | | |
| In hostel/rented house/quarter | 116(50.7) | 113(49.3) | Ref | Ref |
| In own home | 52(34.4) | 99(65.6) | 1.9(1.3–2.9) * | 2.1(1.2–3.6) * |
| **Average monthly income** | | | | |
| Below 40,000 (USD 326) | 119 (52) | 110(48.0) | 0.4(0.3–0.7) *** | 0.7(0.4–1.3) |
| Above 40,000 (USD 326) | 49 (32.5) | 102(67.5) | Ref | Ref |
| **Staying away from your family** | | | | |
| Yes | 101 (48.8) | 106 (51.2) | 1.5(1.0–2.3) * | 0.9(0.6–1.6) |
| No | 67 (38.7) | 106(61.3) | Ref | Ref |
| **Got training/orientation regarding COVID19** | | | | |
| Yes | 48 (30.6) | 109(69.4) | 0.4(0.3–0.6)*** | 0.5(0.3–0.8) * |
| No | 120 (53.8) | 103(46.2) | Ref | Ref |
| **Health Professional** | | | | |
| Radiological | 21 (63.6) | 12 (36.4) | Ref | Ref |
| Doctor | 18 (20.9) | 68 (79.1) | 6.6(2.7–15.9) *** | 3.8 (1.5–10.1)* |
| Medical Laboratory | 31 (47.7) | 34 (52.3) | 1.9 (0.8–4.5) | 2.1(0.8–5.2) |
| Nurse | 48 (51.1) | 46 (48.9) | 1.7 (0.7–3.8) | 1.8 (0.7–4.5) |
| Paramedics | 50 (49) | 52 (51) | 1.8 (0.8–4.1) | 1.3 (0.6–3.2) |
| **Receive the first dose of COVID-19 vaccine** | | | | |
| Yes | 109 (41.1) | 156 (58.9) | Ref | Ref |
| No | 59 (51.3) | 56 (48.7) | 0.6 (0.4–1.0) | 0.7 (0.4–1.1) |

*Significant at p <0.05

** Significant at p <0.01

*** Significant at p <0.001

This might be due to the wide dissemination of correct information regarding the pandemic and disease to the public through various social networks, systems, and support from the community towards HCWs over time. Another reason might be the vaccination program for HCWs, which was started in Nepal immediately after the first wave of the COVID-19 pandemic. This might have also contributed them to experience lower levels of psychological distress.

Regarding gender differences, we found that females had 2.3 times higher odds of perceived stress than male HCWs in Nepal. This is consistent with the findings of the study carried out among healthcare workers in Italy [26], where females had higher perceived stress than males. This finding is also comparable with the outcome of the studies carried out on gender differences in the psychological response to COVID-19 in Nepal [3, 28] and Brazil [29]. The reason can be hypothesized that nurses (only female in Nepal) comprise most healthcare workers with a higher workload, low salary, and greater risk of direct exposure to the infection

increased the stress level. Females also have more responsibility to care for their children and family which might have also contributed to elevating perceived stress.

This study revealed a significant relationship between the profession and perceived stress. Compared to radiological professionals, medical doctors were found to have 0.2 times lower odds of perceived stress. The finding is similar to the study conducted in China during the COVID-19 pandemic [30]; however, few previous studies reported higher stress among medical doctors [3, 31]. The coping mechanisms like training, mediation, health system strengthening to make sure that sufficient human workforce and accepting the problems with courage and fight against the pandemic followed in due course of time might have contributed less to perceived stress among medical doctors. On the other hand, higher stress among radiological personnel might have been attributed to the limited number of radiological personnel who have to do long time duty in limited spaces (rooms) with fear of being infected. Furthermore, being in direct contact with COVID-19 patients, deprived of personal protective measures, and the lack of required training to combat the virus might have put them under more stress than other medical professionals.

Findings from a study conducted in Italy among the general population reported the significant association between income and perceived stress. They justified this result as higher income countries might be closely associated with less concern about the economic burden and lower perceived stress [32]. However, like our study, no significant association was found in a study conducted among HCWs of Italy regarding income and perceived stress level [26].

Healthcare workers are more likely to face stigmatization around the globe during the current pandemic, and infectious diseases, which are potentially deadly conditions and new diseases, often had faced stigma in the past years [33]. From the current study, the healthcare workers from nuclear families reported 1.7 higher odds of being stigmatized than those who belonged to joint/extended families. The reason might be that more members of the family would give opportunities to share different feelings and options to support and tackle them in a better way.

Regarding the location of working institutions, HCWs working in the Bagmati province (the capital region where specialist referral hospitals are located) expressed 3.5 times higher odds of being stigmatized than those working in Sudhurpaschim Province. HCWs had been accused of spreading the coronavirus in the locality and threatening them to vacate their room as soon as possible by the house owners, neighbors even by local leaders and police [34]. This type of community behavior might have contributed to the perceiving higher stigma among HCWs. In addition, most of the participants were from Bagmati Province in our study which might have affected the result. Similarly, HCWs from Karnali Province reported 2.9 higher odds of being stigmatized than the HCWs from Sudhurpaschim Province. Lack of proper knowledge and awareness of the COVID-19 pandemic might have increased stigmatization as Karnali province is the most remote, having fewer health facilities.

This study illustrated that HCWs from government hospitals had a lower chance of feeling stigmatized than those working at the COVID-19 fever clinic. The reason might be explained as government hospitals have already set up a structure with fine management of human resources. They might have a well-adjusted health system, and the support from the government makes them feel confident. On the other hand, the fever clinics are recently set up to tackle the pandemic with temporary staffing. This might increase fear of uncertainty and chances of being more stigmatized. Furthermore, the fever clinic was the primary point of contact for the COVID-19 patients, so, the majority of the infected patients visited. Because of their close contact with COVID-19 patients and the potential for disease transmission, health staff working there may have been ostracized even more.

Social support is crucial to mitigate mental health burdens and maintaining well-being in the HCWs. Its importance is invaluable during the public health emergencies like the COVID-19 pandemic. Various studies have supported the fact that higher social support is associated with better mental well-being during the COVID-19 pandemic [13, 35]. The HCWs in this study reported higher odds of perceiving social support who were staying at their own house. It is anticipated fear of being infected lies to other members when living in rented houses. Thus, it can be concluded that living in own house might have strong social support. But surprisingly, this study showed; lower social support among HCWs who were oriented or trained about COVID-19 compared to those who were not trained about COVID-19, respectively.

This study included representatives from all health care professionals working on the front line during the pandemic. To the best of our knowledge, this is one of the early papers assessing the actual frequency of stress, stigma, and social support among health care workers during the early pandemic in Nepal. Despite its significant importance in evidence, our study had several limitations that should be considered when interpreting the data. All the measurements in this study were based on self-reports, which may have been prone to response and information bias. Contextualization of questionnaire items and validation of the Nepali version of stigma and social support seem necessary if used to assess stigma and social support in the Nepali population in the future. This study is cross-sectional and has predictive limitations as exposure and outcome were assessed at once. We are unable to discriminate preexisting psychological distress from newly emerging symptoms, and also, we are unable to assess the secondary stressors such as personal and relationships, which may have an impact on outcome variables, which were not measured. As this was a web-based study, limited access to the internet may have discouraged the HCWs from participating in the survey. However, at that time it was the most practicable method of assessing information from health care workers. The sample is not representative because of a non-random sampling technique, which may be a drawback of our study; nevertheless, a random sampling technique can be used to do subsequent research. However, it was one of the most practical ways of conducting research and demonstrating the reality at that time.

## Conclusion

The prevalence of perceived stress was high among female, and health care workers currently working in Karnali province. Likewise, the magnitude of perceived stigma in this study was low compared to the studies done in the early phase of the pandemic in Nepal. Also, fifty percent of the HCWs revealed moderate social support. Still, the present finding among the study population highlights the need for the provision of immediate psychosocial care. Similarly, it is important to conduct a mass awareness campaign to ameliorate public knowledge of the stigma and discrimination associated with the pandemic. Good provision of quarter/residence to HCWs will help to reduce the stigma they face at their residence and neighborhood as many incidents have been reported, like forcing HCWs to vacate their rented room. Training/orientation programs on managing health emergencies would be better for the new HCWs recently recruited in the COVID-19 dedicated hospital or fever clinic. Finally, this study suggests formulating and implementing appropriate plans and policies to mitigate the distress in the context of the COVID-19 pandemic and for a part of preparedness in the future. From this pandemic, we have learned that health information plays a vital role as a tool of education to combat future health emergencies. This should be user-driven, with proper allocation of resources easy to understand and accessible at the point of use. In addition to this, a good understanding of local disease patterns would help in formulating strategies so that the burden on healthcare workers would be less during emergencies.

## Supporting information

**S1 Table. Association between stress with independent variables.**
(DOCX)

**S2 Table. Sociodemographic factors associated with perceived stigma.**
(DOCX)

**S3 Table. Association between sociodemographic factors and social support.**
(DOCX)

## Author Contributions

**Conceptualization:** Lok Mani Giri, Kiran Paudel, Gabriel Gulis.

**Data curation:** Lok Mani Giri, Kiran Paudel, Tara Ballav Adhikari.

**Formal analysis:** Lok Mani Giri, Kiran Paudel, Tara Ballav Adhikari, Gabriel Gulis.

**Investigation:** Lok Mani Giri, Gabriel Gulis.

**Methodology:** Lok Mani Giri, Kiran Paudel, Sandesh Bhusal, Gabriel Gulis.

**Project administration:** Lok Mani Giri, Gabriel Gulis.

**Resources:** Lok Mani Giri, Gabriel Gulis.

**Software:** Lok Mani Giri, Kiran Paudel.

**Supervision:** Lok Mani Giri, Gabriel Gulis.

**Validation:** Lok Mani Giri.

**Visualization:** Lok Mani Giri, Sandesh Bhusal.

**Writing – original draft:** Lok Mani Giri, Kiran Paudel, Sandesh Bhusal, Tara Ballav Adhikari, Gabriel Gulis.

**Writing – review & editing:** Lok Mani Giri, Kiran Paudel, Sandesh Bhusal, Tara Ballav Adhikari, Gabriel Gulis.

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
