## [Decision Letter · Decision Letter 0]

3 Jan 2022

PGPH-D-21-00826

Perceived stress, stigma and social support among Nepalese health care workers during COVID-19 Pandemic: a cross-sectional web-based survey

Dear Dr. Paudel,

Thank you for submitting your manuscript to PLOS Global Public Health. After careful consideration, we feel that it has merit but does not fully meet PLOS Global Public Health’s publication criteria as it currently stands. Therefore, we invite you to submit a revised version of the manuscript that addresses the points raised during the review process.

We look forward to receiving your revised manuscript.

Kind regards,

Behdin Nowrouzi-Kia

Academic Editor

Journal Requirements:

1. Please provide additional details regarding participant consent. In the ethics statement in the Methods and online submission information, please ensure that you have specified whether consent was informed.

2. Please amend your Data Availability Statement and indicate where the data may be found.

Additional Editor Comments (if provided):

Reviewers' comments:

Reviewer's Responses to Questions

**Comments to the Author**

1. Does this manuscript meet PLOS Global Public Health’s publication criteria? Is the manuscript technically sound, and do the data support the conclusions? The manuscript must describe methodologically and ethically rigorous research with conclusions that are appropriately drawn based on the data presented.

Reviewer #1: Partly

Reviewer #2: Yes

2. Has the statistical analysis been performed appropriately and rigorously?

Reviewer #1: No

Reviewer #2: Yes

3. Have the authors made all data underlying the findings in their manuscript fully available (please refer to the Data Availability Statement at the start of the manuscript PDF file)?

Reviewer #1: Yes

Reviewer #2: No

4. Is the manuscript presented in an intelligible fashion and written in standard English?

Reviewer #1: No

Reviewer #2: Yes

5. Review Comments to the Author

Reviewer #1: It is well known that HCW are subjected to high stress and stigma as well as with poor social support during the COVID-19 pandemic. There is one study conducted in Nepal itself and published in PLOS ONE. Similar studies were conducted in several other nations as well (to name a few India, China, Jordan, Egypt, Italy, USA etc.). Even there are three review studies published on the same topic. Hence what this study adds to the existing evidence? The introduction lacks justification of conducting this study.

The sample size computation was based on estimation of a proportion. With an anticipated proportion of 0.56 and absolute precision of 5%, the minimum sample size required for 95 % confidence level is 379. Incorporating 10% non-response rate, the final sample size is 422. However, the authors mentioned the final sample size as 404 which is not correct.

The mean perceived stress score (PSS) is considered as the cut-off for defining the low and high stress. This is not a good practice to dichotomize a continuous variable based on location parameters like mean or median. There are several articles published that discussed about the danger of use this practice.

The MERS-CoV cut-off used was 26. What was the basis of using this cut-off?

In page no. 6 line no. 114, “After excluding the 24 samples …”, the use of “24 samples” is wrong. Instead, it should be “24 participants”.

Those variables which were significant in univariate analysis were only included in multiple logistic regression. This is not a good practice as there may be some non-significant variables with P value close to 0.05 which may be clinically important or may become significant in multiple logistic regression. The authors should have used P<0.2 or P<0.25 as a cut-off for inclusion in multivariable analysis instead of P<0.05 obtained in univariate analyses.

Use of the term “multivariate” is wrong instead it should be “multivariable”.

In Table 3 the perceived stress was categorized in to low, moderate and high. But only one cut-off (based on the mean score of 26) was used. So how three categories were formed?

There were 191 participants with high perceived stress (based on Table 4) which constitute 50.3% of the total sample size. Since this is a cross sectional study, the appropriate summary measure is prevalence ratio (PR) instead of odds ratio (OR). When the event rate is high (more than 10%), the OR obtained using logistic regression over estimates the PR. In such cases modified Poisson regression with robust variance estimation is the suggested analysis which gives the PR and corresponding 95% CI’s.

In Table 4 the unadjusted OR for average income<40000 is 1.6 but the adjusted OR is 0.7. This may be because of multicollinearity. It is better to check this.

In Tables 4 & 5, a few OR’s are with very low values like 0.5 or 0.4 but still not significant. This could be mainly due to small sample size in the respective categories. The authors should consider this while interpreting the results as well as in drawing the conclusions.

Interpreting OR as risk is wrong as OR compares the odds of an event between two groups and not the risk.

One of the main limitations of this study is that the sample is not a representative sample as the sampling technique used was non-random sampling. This should be stated in the limitations part.

Some of the latest similar studies were not referred in the manuscript.

Reviewer #2: Thank you for a very fascinating paper. I am making the following suggestions in the hopes that it will strengthen your work further

1. Introduction: can you reconfigure your introduction to comment on the Nepalese ( or general public's) perception of doctors in usual times ie are they typically respected, stigmatized based on specific patient populations they work with and if so what are the known effects of these in Nepal ie do people avoid certain fields of practice? are there known mental health issues? Then I would talk about mental health effects of COVID globally and why your study was conceived ie how are these different in LMICÉ in a country where stigma may or may not exist according to area of practice pre-pandemic etc...and then why is it important to know your study results

2. Methods: why did you use Oman? Please add detail here. Please add detail of whether you asked about any perceived stigma before the pandemic ? Did you resend the survey to get responses? what happened with incomplete surveys ie missing data? How were surveys anonymized-- can you provide more detail of how written consent was separated from data collection? In demographics was vaccine considered one or two doses?

3. Results- please give a sense of what income values means in Nepal context ie what is considered poverty line and/or what is equivalent in USD

- Could you please clarify what you mean that the highest levels of stress were perceived among professions ( do you mean professionals?) and that doctors experienced the highest stress? From the table it looks like the physician group was least stressed, am I correct? And that nurses were highest?

- please explain what nuclear family vs joint or extended family means -- maybe in methods section when describe data collected. I think this is very important re understanding who is family in Nepal and its implications. Can you comment specifically in the text on the issue of gender here please

4. Discussion

- Can you comment on the role of education re pandemic and differences if any received by doctors and nurses on stress levels?

- you mention coping results in lowered stress in physician - what coping are you referring to?

- can you comment on how greater exposure to COVID patients decreased stress levels in other studies and whether you feel your data is aligned with this result? this may be why radiologists are overall more stressed?

- Please address the reasons why stigma may be higher in some provinces , is it education related? based in fear? more cases per capita? could it be mitigated in future? Because if HCWs were chased from homes and the community this is not perceived stigma its real-- there is a difference here. Did others have to leave their dwellings too? do you have that data? What or how did the nature of family change perceptions of stigma?

- I am not sure the perception of stigma is dissipated because of social supports ( which I would think would help with stress but not the perception of stigma?) or whether its just not perceived as strongly because people are already in the family relationships expected of them ie married etc. Can you discuss this please? In other words if you are already different because you are single is the increased stigmatization felt more acutely?

- same issue with hospitals-- feeling stigmatized is how others perceive you-- its not a self-perception thing. So confidence should not play any role ie public hospitals. The finding that doctors in a fever hospital felt more stigmatized would likely therefore be more due to the public having a sense ( real or not) that these doctors were seeing more COVID patients because COVID is known to cause fever.. therefore these doctors posed a higher risk to public. If doctor was working in a public hospital the public doesn't "know" in the same way that they are seeing a lot of COVID patients.....

- lving in rented quarters would have been done to protect family would it not? So would the different results be explained that the rented quarters were filled with other HCWs or that HCWs were mostly living alone when renting? Please clarify

- similarly lower supports were provided to people who received more COVID training-- is this because even family members stood back because they equated receiving training with more exposure to COVID patients?

- If there would there be any value in providing more robust public and family (of HCWs) education to help HCWs in the future, how and what would need to be explained? The provision of good living quarters would decrease family support and may result in stigmatization of whole neighbourhoods so I would clarify this statement

6. PLOS authors have the option to publish the peer review history of their article (what does this mean?). If published, this will include your full peer review and any attached files.

**Do you want your identity to be public for this peer review?** For information about this choice, including consent withdrawal, please see our Privacy Policy.

Reviewer #1: No

Reviewer #2: No

---

## [Decision Letter · Decision Letter 1]

22 Mar 2022

PGPH-D-21-00826R1

Perceived stress, stigma and social support among Nepalese health care workers during COVID-19 Pandemic: a cross-sectional web-based survey

Dear Dr. Paudel,

Thank you for submitting your manuscript to PLOS Global Public Health. After careful consideration, we feel that it has merit but does not fully meet PLOS Global Public Health’s publication criteria as it currently stands. Therefore, we invite you to submit a revised version of the manuscript that addresses the points raised during the review process.

We look forward to receiving your revised manuscript.

Kind regards,

Behdin Nowrouzi-Kia

Academic Editor

Journal Requirements:

1. Please ensure that you refer to Table 1 in your text as, if accepted, production will need this reference to link the reader to the table.

2. We notice that your supplementary tables are included in the manuscript file. Please remove them and upload them  with the file type 'Supporting Information'. Please ensure that all Supporting Information files are included correctly and that each one has a legend listed in the manuscript after the references list.

Additional Editor Comments (if provided):

Reviewers' comments:

Reviewer's Responses to Questions

**Comments to the Author**

1. If the authors have adequately addressed your comments raised in a previous round of review and you feel that this manuscript is now acceptable for publication, you may indicate that here to bypass the “Comments to the Author” section, enter your conflict of interest statement in the “Confidential to Editor” section, and submit your "Accept" recommendation.

Reviewer #2: All comments have been addressed

Reviewer #3: All comments have been addressed

2. Does this manuscript meet PLOS Global Public Health’s publication criteria? Is the manuscript technically sound, and do the data support the conclusions? The manuscript must describe methodologically and ethically rigorous research with conclusions that are appropriately drawn based on the data presented.

Reviewer #2: Yes

Reviewer #3: Yes

3. Has the statistical analysis been performed appropriately and rigorously?

Reviewer #2: Yes

Reviewer #3: Yes

4. Have the authors made all data underlying the findings in their manuscript fully available (please refer to the Data Availability Statement at the start of the manuscript PDF file)?

Reviewer #2: Yes

Reviewer #3: Yes

5. Is the manuscript presented in an intelligible fashion and written in standard English?

Reviewer #2: Yes

Reviewer #3: Yes

6. Review Comments to the Author

Reviewer #2: thank you for addressing all my comments

Reviewer #3: Please see attached file.

7. PLOS authors have the option to publish the peer review history of their article (what does this mean?). If published, this will include your full peer review and any attached files.

**Do you want your identity to be public for this peer review?** For information about this choice, including consent withdrawal, please see our Privacy Policy.

Reviewer #2: No

Reviewer #3: No

---

## [Decision Letter · Decision Letter 2]

14 Apr 2022

Perceived stress, stigma, and social support among Nepali health care workers during COVID-19 pandemic: a cross-sectional web-based survey

PGPH-D-21-00826R2

Dear Mr. Paudel,

We are pleased to inform you that your manuscript 'Perceived stress, stigma, and social support among Nepali health care workers during COVID-19 pandemic: a cross-sectional web-based survey' has been provisionally accepted for publication in PLOS Global Public Health.

Best regards,

Behdin Nowrouzi-Kia

Academic Editor

Reviewer Comments (if any, and for reference):

Reviewer's Responses to Questions

**Comments to the Author**

1. If the authors have adequately addressed your comments raised in a previous round of review and you feel that this manuscript is now acceptable for publication, you may indicate that here to bypass the “Comments to the Author” section, enter your conflict of interest statement in the “Confidential to Editor” section, and submit your "Accept" recommendation.

Reviewer #3: All comments have been addressed

2. Does this manuscript meet PLOS Global Public Health’s publication criteria? Is the manuscript technically sound, and do the data support the conclusions? The manuscript must describe methodologically and ethically rigorous research with conclusions that are appropriately drawn based on the data presented.

Reviewer #3: Yes

3. Has the statistical analysis been performed appropriately and rigorously?

Reviewer #3: Yes

4. Have the authors made all data underlying the findings in their manuscript fully available (please refer to the Data Availability Statement at the start of the manuscript PDF file)?

Reviewer #3: Yes

5. Is the manuscript presented in an intelligible fashion and written in standard English?

Reviewer #3: Yes

6. Review Comments to the Author

Reviewer #3: The authors have satisfactorily addressed all prior comments.

7. PLOS authors have the option to publish the peer review history of their article (what does this mean?). If published, this will include your full peer review and any attached files.

**Do you want your identity to be public for this peer review?** For information about this choice, including consent withdrawal, please see our Privacy Policy.

Reviewer #3: No
